# Incidence of Severe COVID-19 Outcomes and Immunization Rates in Apulian Individuals with Inflammatory Bowel Disease: A Retrospective Cohort Study

**DOI:** 10.3390/vaccines12080881

**Published:** 2024-08-02

**Authors:** Francesco Paolo Bianchi, Antonella Contaldo, Maurizio Gaetano Polignano, Antonio Pisani

**Affiliations:** 1Health Prevention Department, Local Health Authority of Brindisi, 72100 Brindisi, Italy; 2National Institute of Gastroenterology, IRCCS S. De Bellis, Research Hospital, 70013 Castellana Grotte, Italymaurizio.polignano@irccsdebellis.it (M.G.P.); antonio.pisani@irccsdebellis.it (A.P.)

**Keywords:** Crohn’s disease, ulcerative colitis, vaccine compliance, SARS-CoV-2, gastroenterology, epidemiology

## Abstract

The etiology of Inflammatory Bowel Disease (IBD) is not fully understood but is believed to involve a dysregulated immune response to intestinal microbiota in genetically susceptible individuals. Individuals with IBD are at increased risk of infections due to immunosuppressive treatments, comorbidities, and advanced age. Current evidence indicates that IBD patients are not at higher risk of SARS-CoV-2 infection compared to the general population, though the risk of severe outcomes remains debated. A retrospective observational study was conducted using Apulian regional health data from 2020 to 2022. This study included 1029 IBD patients and 3075 controls, matched by age and sex. COVID-19 incidence, hospitalization, and case fatality rates were analyzed alongside vaccination coverage. No significant differences in COVID-19 incidence (IRR = 0.97), hospitalization (*p* = 0.218), or lethality (*p* = 0.271) were evidenced between IBD patients and the general population. Vaccination rates were high in both groups, with slightly higher uptake in IBD patients. Multivariate analysis identified age and male sex as risk factors for severe COVID-19 outcomes, while vaccination significantly reduced hospitalization and lethality risks. IBD patients in Apulia do not have an increased risk of COVID-19 infection or severe outcomes compared to the general population. Vaccination is crucial in protecting IBD patients, and ongoing efforts to promote vaccination within this population are essential. Future research should focus on the impact of specific IBD treatments on COVID-19 outcomes and the long-term effectiveness of vaccines.

## 1. Introduction

Inflammatory Bowel Disease (IBD) refers to ongoing inflammation in the digestive tract, primarily manifesting as Crohn’s disease and ulcerative colitis, which are its most common forms [1]. The epidemiology of Inflammatory Bowel Disease (IBD) is rapidly evolving worldwide. The estimated prevalence (over 0.3%) continues to rise in Western countries, particularly in North America, Oceania, and Europe, where the burden of IBD is substantial [2]. In Italy, the Global Burden of Diseases (GBD) study reported an IBD prevalence of 80.9 per 100,000 population in 1990, accounting for 56,469 cases. By 2017, this figure had risen to 93.8 per 100,000 population, indicating a significant increase in IBD cases over the past few decades [3].

While the precise cause of IBD is not fully understood, it is generally accepted that a misdirected immune response against the intestinal microbiota, combined with genetic predisposition, plays a significant role in its development [4]. Despite ongoing research into this intricate relationship, it is clear that individuals with IBD are at a higher risk for complications from infectious diseases. This increased risk is due to several factors, including the use of immunosuppressive medications to control inflammation, a greater likelihood of having additional chronic conditions compared to those without IBD, and the immunocompromised state often associated with older age [5,6,7,8].

In this scenario, SARS-CoV-2 attaches to its target cells via the angiotensin-converting enzyme 2 (ACE2), which is naturally found in epithelial cells of the lungs, intestines, kidneys, and blood vessels. The terminal ileum and colon have some of the highest concentrations of ACE2 in the body. Importantly, ACE2 expression is elevated in the inflamed intestines of individuals with IBD [9]. Moreover, the SARS-CoV-2 virus requires a specific fusion or ‘spike’ protein to merge with host cell membranes, a key step for infection. This spike protein is activated by host cell trypsin-like proteases, which are reported to be upregulated in IBD [9]. These findings imply that the inflamed intestines of IBD patients might act as a favorable entry point for the virus into human tissues.

Nevertheless, the scientific literature agrees that individuals with IBD are not at a higher risk of SARS-CoV-2 infection than the general population [9,10,11]. On the other hand, less clear is the risk for hospitalization and death in these individuals compared to the general population. Xu Fang et al. [12] investigated US subjects aged ≥67 years in the first half of 2020 (April–July), finding an increased risk for hospitalization among individuals with IBD, while the risk of death was similar compared to the general population. A 2021 Swedish survey conducted between February and July 2020 found that people with IBD had a higher likelihood of being hospitalized due to COVID-19 compared to the general population. However, when compared to their siblings, individuals with IBD did not show an increased risk of hospitalization. Additionally, the study found no heightened risk of severe COVID-19 outcomes, such as needing intensive care or death from the virus [13]. Allocca M et al. [14] concluded a prospective case series of 41 individuals with IBD and concomitant COVID-19 infection, showing that these patients were not at increased risk of worse outcomes from COVID-19. Similar results were obtained on a cohort of 83 US patients [15]. A 2020 Italian study conducted using 79 patients reported a significant risk of COVID-19 pneumonia and death in patients with active disease [16].

In light of the available yet not entirely conclusive evidence, international gastroenterology scientific societies and public health organizations advocate for COVID-19 vaccination in patients with IBD and those who are immunocompromised [17,18]. Even though phase III clinical trials for COVID-19 vaccines did not encompass IBD patients or individuals on immunosuppressive medications, subsequent phase IV studies have shown that the vaccines are highly effective, capable of inducing an immune response, and safe for this particular group [19,20,21,22]. In Italy, from the onset of the COVID-19 vaccination campaign, individuals with IBD have never been explicitly mentioned as a high-risk group. Instead, the guidelines have consistently referred to immunocompromised individuals, specifically those undergoing immunosuppressive therapy [23]. This classification has remained unchanged even for subsequent booster doses, continuing to encompass immunocompromised patients due to their treatments without directly naming those with IBD [24].

In this context, our study aims to compare the incidence of COVID-19 and its outcomes, specifically hospitalization and case fatality rates, between a cohort of patients with IBD and the general population in the Apulia region (southern Italy, 4,000,000 inhabitants). Additionally, we will focus on the role of COVID-19 vaccines in preventing these outcomes and on the determinants of vaccination compliance in the IBD population. By analyzing these parameters, we seek to understand whether individuals with IBD are at a higher risk of contracting COVID-19 and experiencing severe disease outcomes compared to the general population and how vaccination status influences these risks. This comparison will help to inform clinical practices and public health policies, ensuring they are tailored to protect and manage this high-risk group more effectively.

## 2. Materials and Methods

This study is a retrospective observational analysis. The study population was identified using the Apulian Regional Archive of Hospital Discharge Forms (SDO), an online database that contains comprehensive information on hospital and inpatient procedures across the region [25]. We included all records related to ICD9 codes for Crohn’s disease (55.5x) and ulcerative colitis (55.6x), focusing on diagnoses made between 2020 and 2022 at the National Institute of Gastroenterology, Research Hospital “De Bellis” in Castellana Grotte, Apulia. Only residents of Apulia were included. Two epidemiologists with expertise in health information independently reviewed all records to reduce the risk of misclassification and incorrect diagnosis. They agreed that none of the records posed a high risk of misclassification, so all records were included in the final data analysis. Cases of IBD were then matched with Apulian residents without an IBD diagnosis at a 1:3 ratio to strengthen the statistical analysis. The control group was selected using identical data records, criteria, and time frames (2020–2022) as the IBD group. Group assignments were randomized, ensuring homogeneity for age and sex at the start of the pandemic. Randomization was conducted using STATA MP18 software.

The COVID-19 vaccination status was determined using the Regional Immunization Database (GIAVA) and/or the Italian COVID-19 Immunization Database [12]. Data on COVID-19 cases recorded from March 2020 to April 2023 were extracted from the Italian Institute of Health (ISS) platform, “Integrated Surveillance of COVID-19 Cases in Italy.” This surveillance includes all COVID-19 cases diagnosed by regional and national reference laboratories, as well as COVID-19-related hospitalizations and deaths [25].

To enhance our sample characterization, we cross-referenced information on chronic diseases using the Edotto platform, identifying user-fee exemption codes. These data were further integrated with information from the ISS platform and the archive of hospital discharge forms [25].

Data sources were extracted and matched using patients’ unique identification numbers (PINs). We excluded foreign subjects with a temporary unique identification number, as they might have returned to their countries or settled in Italy, obtaining a non-temporary PIN that we could not trace. Our study concluded in April 2023.

The final dataset was created as an Excel spreadsheet that included the group variable (individual with IBD vs. general population), sex, age at the start of the COVID-19 pandemic, chronic disease(s), IBD disease (Crohn/ulcerative colitis), diagnosis of COVID-19 (YES/NO), vaccine prophylaxis (YES/NO), COVID-19-related hospitalization (YES/NO), and death (YES/NO). Anonymized data analysis was performed using STATA MP18 software.

Continuous variables are presented as mean ± standard deviation and range, while categorical variables are expressed as proportions, with 95% confidence intervals (95%CIs) provided when applicable. Proportions between groups were compared using the chi-square or Fisher’s exact tests. The normality of continuous variables was assessed using the skewness and kurtosis test; Student’s *t*-test for independent samples was employed to compare continuous variables between groups.

Vaccine coverages (VCs) were defined as follows:-Baseline routine: Subjects vaccinated with two doses of the BNT162b2 mRNA vaccine, two doses of the mRNA-1273 vaccine, two doses of the ChAdOx1-S vaccine, one dose of the Ad26.COV2.S vaccine, or a mixed schedule; this includes subjects who were still alive at the start of the vaccination campaign in January 2021.-First booster dose: Subjects who received the first booster dose, compared to subjects still alive in January 2022.-Second booster dose: Subjects who received the second booster dose, compared to subjects still alive at the start of the second booster vaccination campaign in April 2022.

Overall survival (OS) was defined as the time elapsed from the start of the COVID-19 pandemic to the diagnosis of COVID-19 or the end of the observation period. OS was assessed using Kaplan–Meier curves, and the differences between groups were evaluated using the log-rank test. The COVID-19 incidence rates (IRs) (×100 persons-month), defined as the number of new cases of COVID-19 that occurred in the studied population within a defined period of time, the proportion of hospitalizations, and the case fatality rates were estimated. The incidence rate ratio (IRR), calculated by dividing the IR of individuals with IBD by the IR of the general population, was estimated to compare the IRs between groups, indicating 95%CIs. To analyze the predictors of OS, a multivariate Cox semiparametric regression was constructed using sex (male vs. female), age (years), chronic diseases, and COVID-19 baseline routine (YES/NO) as determinants. The adjusted hazard ratio (aHR) with 95% CI was calculated. The proportionality assumption of the multivariate Cox semiparametric regression model was assessed using the Schoenfeld and scaled Schoenfeld residuals tests; variables not meeting proportionality were excluded. The Gronnesby and Borgan test was used to evaluate the model’s goodness of fit.

To analyze the determinants of COVID-19 vaccine uptake (basal routine, first booster, and second booster), three multivariate logistic regression models were built; the group variable (individuals with IBD vs. general population, Appendix A) was considered as the main determinant, adjusted for sex (male vs. female), age (years), and chronic diseases. These models have been repeated using the comparison between Crohn’s disease and ulcerative colitis as the group variable. To analyze the determinants of COVID-19 hospitalization and death (YES/NO), two multivariate logistic regression models were built; group variable (individuals with IBD vs. general population) was considered as the main determinant, adjusted for sex (male vs. female), age (years), chronic diseases, and COVID-19 basal routine (YES/NO). For all models, the adjusted odds ratios (aORs) were calculated, as well as 95%CIs. Hosmer–Lemeshow’s chi-squared test was used to evaluate the goodness-of-fit of multivariate logistic regression models.

A two-sided *p*-value < 0.05 was considered an indicator of statistical significance for all tests.

The research was carried out in accordance with the principles of the Helsinki Declaration; as this study constituted public health surveillance, ethical approval from the institutional review board was not required.

## 3. Results

Our sample consisted of 4140 subjects, of which 1029 (25.1%) were affected by IBD and 3075 (74.9%) belonged to the control group. Table 1 reports the characteristics of our sample per group.

The IR of SARS-CoV-2 infection was 1.34 (95%CI = 1.28–1.40) per 100 persons-month (*n* = 1,751), without a statistically significant difference between individual with IBD (IR: 1.31; 95%CI = 1.18–1.44; *n* = 431) and the general population (IR: 1.35; 95%CI = 1.28–1.43; *n* = 1320), with an IRR of 0.97 (95%CI = 0.86–1.08; *p*-value = 0.484). The cumulative incidence of infection in the two groups (log-rank *p*-value = 0.484) is shown in Figure 1. No differences were observed when stratifying IRs per age, class, and sex (Appendix A).

The multivariate semiparametric Cox regression confirmed that IBD was not a predictor for COVID-19 infection (aHR = 0.95; 95%CI = 0.84–1.07; Appendix A).

The proportion of hospitalization was 2.0% (*n* = 83), with a case fatality rate of 0.2% (*n* = 10), without differences between groups in both cases [hospitalization rate: general population= 2.2% (*n* = 67) vs. individuals with IBD = 1.6% (*n* = 16); *p*-value = 0.218. Case fatality rate: general population = 0.3% (*n* = 9) vs. individuals with IBD = 0.1% (*n* = 1); *p*-value = 0.271].

No differences in IRs, hospitalization, and case fatality rate were evidenced considering the diseases (Appendix A). The characteristics of hospitalized and dead individuals with IBD, per disease, are reported in Appendix A.

Table 2 describes the COVID-19 vaccine coverage; considering the basal routine and booster doses, no statistically significant differences (*p* > 0.05) are evident between IBD diseases (Appendix A).

The multivariate regression models evidence that older age is associated with better vaccine uptake, whereas the presence of comorbidities significantly influences booster dose compliance. No significant differences in vaccination behavior were observed between individuals with IBD and the general population (Table 3). Similar conclusions can be drawn when focusing exclusively on the IBD population, where specific disease characteristics do not appear to impact vaccination status (Appendix A).

The multivariate analyses of the determinants of COVID-19-related hospitalization and death are described in Table 4.

## 4. Discussion

Our study has revealed no significant differences in the incidence of COVID-19, the risk of hospitalization, or mortality rates between individuals with IBD and the general population in the Apulia region. This finding is consistent across different types of IBD, including Crohn’s disease and ulcerative colitis, indicating that the presence of IBD itself does not confer additional risk for these specific COVID-19 outcomes when compared to the broader population. Our results align with the evidence reported in a 2022 meta-analysis [26] investigating studies published from 2020 to 2021. This meta-analysis concluded that IBD was not significantly associated with an increased risk of COVID-19 infection, hospitalization, severe COVID-19, or mortality. Compared to other high-risk patient categories, individuals with IBD appear to be less susceptible to severe outcomes of COVID-19. A 2023 study on Apulian splenectomized patients indicated that splenectomy was not associated with an increased risk of COVID-19 infection. However, the proportions of hospitalization and case fatality rates were significantly higher compared to the Apulian general population (2.9% vs. 0.5% and 2.6% vs. 0.4%, respectively), especially among older individuals. The higher risk of COVID-19-related hospitalizations and death in splenectomized patients was confirmed via multivariate regression analysis (adjusted odds ratio >1) [25].

Furthermore, the analysis has identified that demographic factors such as advanced age and male sex are associated with an increased risk of severe COVID-19 complications. These findings align with the existing literature, which consistently reports that older adults and males are more susceptible to severe outcomes from COVID-19 [27,28]. This reinforces the need for heightened vigilance and protective measures for these demographics, regardless of their IBD status.

An encouraging observation from our study is that COVID-19 vaccination coverage among IBD patients is slightly higher than that in the general population. The values in our population are indeed higher than the vaccine coverages reported in a 2023 meta-analysis on a global scale. The authors of the meta-analysis reported a pooled prevalence of COVID-19 vaccine uptake of 72% (95%CI = 59–83%) for at least one dose, 81% (95%CI = 68–91%) for the complete vaccination regimen, and 71% (95%CI = 46–91%) for the third dose [29]. This increased vaccination rate can be attributed to targeted vaccination campaigns that have progressively focused on chronic disease patients over the years. However, despite these reassuring findings, it is critical to acknowledge that the absence of significant differences in COVID-19 incidence and outcomes between IBD patients and the general population does not imply that IBD patients are risk-free; indeed, up to 7% of patients with IBD infected with COVID-19 suffered an IBD flare 3-months post-infection [11]. Moreover, it is essential to note a significant issue regarding the low coverage of booster doses, particularly the second booster. This challenge is well-documented among at-risk populations [25,30] and the general population [31]. This presents a notable public health concern, as booster doses are critical for maintaining long-term immunity, especially in vulnerable groups such as those with chronic conditions like IBD. Indeed, our multivariate regression analysis highlights that vaccination remains a crucial protective factor against severe COVID-19 complications. This underscores the necessity for ongoing efforts to promote vaccination within the IBD community. As evidenced by Bianchi FP et al. [32] in a 2024 meta-analysis, healthcare providers, especially gastroenterologists, play a significant role in emphasizing the importance of immunization for this vulnerable group. This proactive approach has likely contributed to better vaccine uptake among IBD patients, protecting them against severe COVID-19 outcomes.

A strength of our study is that it allows for an evaluation over an extended period (3 years). In contrast, most studies on this topic have focused primarily on the initial months of 2020, with small sample sizes [9,10,11,12,13,14,15,16]. This broader temporal perspective enables us to assess the phenomenon over a longer time period and make more comprehensive medium-term evaluations. By capturing data across a more extensive timeframe, our study can provide insights into the evolving nature of COVID-19 and its impact on individuals with IBD beyond the immediate aftermath of the pandemic’s onset. Another strength is the comparison with the Apulian general population. We estimated this subgroup population’s risk of COVID-19-related hospitalization and death; moreover, we did not only focus on the burden of disease but also on the immunization status. However, the primary limitation of our study is that the data sources do not allow for the assessment of specific therapies taken by the IBD population. The scientific literature indicates that the type of therapy is a crucial determinant of severe COVID-19 outcomes [33,34]. Understanding the impact of various immunosuppressive and biological treatments on COVID-19 severity is vital for providing tailored care to IBD patients. Despite this limitation, it is reasonable to assume that the patients included in our study are under the care of healthcare facilities and, therefore, likely to receive appropriate therapies. Nonetheless, the lack of detailed therapeutic data means our analysis cannot directly correlate treatment types with COVID-19 outcomes. Moreover, some of our data sources are built for administrative and non-epidemiological purposes, so there is a theoretical risk of bias. Moreover, a significant limitation of our study is that the sample size is not large enough for outcomes such as hospitalization and death. This constraint prevents us from drawing robust conclusions about these outcomes. Unfortunately, we are unable to expand the sample size, and we have acknowledged this as a limitation of our analysis. Another area for improvement in our study is the inability to conduct in-depth analyses comparing outcomes of Crohn’s disease and ulcerative colitis. The number of COVID-19 outcomes within these subgroups was too small to draw generalizable conclusions. This limitation prevents us from identifying potential differences in COVID-19 risks and outcomes specific to each type of IBD, highlighting the need for further research with larger sample sizes to assess these distinctions adequately. Furthermore, we could not evaluate the correlation between VCs and community care determinants.

Future scientific studies should address several key areas to gain a more comprehensive understanding of COVID-19 outcomes in patients with IBD, as shown in other immune-mediated inflammatory diseases [35]. Firstly, incorporating detailed information on the specific therapies administered to IBD patients is crucial. Understanding the differential impact of various treatment regimens can help tailor medical care to mitigate risks associated with COVID-19 in this vulnerable population. Another critical focus should be the evaluation of vaccine effectiveness in preventing severe COVID-19 complications among IBD patients. Future studies should explore the long-term protective effects of vaccines, including the impact of booster doses and vaccine-induced immunity over time. Finally, future studies should consider larger, multicenter cohorts to increase the generalizability of findings.

In conclusion, while our study demonstrates that IBD patients in Apulia do not face a higher risk of COVID-19 infection or severe outcomes compared to the general population, the protective role of vaccination must be underlined. Vaccination remains a critical measure in mitigating the risk of severe disease and ensuring the health and safety of IBD patients during the ongoing pandemic. As reported in the literature, immunization in subgroups of the population with chronic conditions is crucial to protect them [36,37].

The involvement of public health institutions is vital in orchestrating effective vaccination campaigns. These campaigns should be strategically designed to reach IBD patients and other high-risk populations, ensuring they receive timely and accurate information about the importance of COVID-19 vaccination. The collaboration between public health authorities and healthcare providers can enhance the outreach and effectiveness of these campaigns [29,32]. The public health sector’s role extends to facilitating seamless interaction between healthcare providers and patients. By creating systems that support efficient vaccine distribution and administration, public health initiatives can ensure that vaccines are readily accessible to those who need them most. This includes maintaining robust records of immunization status and ensuring follow-up for booster doses when necessary. The efforts of gastroenterologists and other healthcare professionals in advocating for COVID-19 vaccination among IBD patients are essential. Their role in educating patients about the benefits of vaccination and addressing any concerns or misconceptions is crucial in achieving high vaccination coverage within this vulnerable group.

## Figures and Tables

**Figure 1 vaccines-12-00881-f001:**
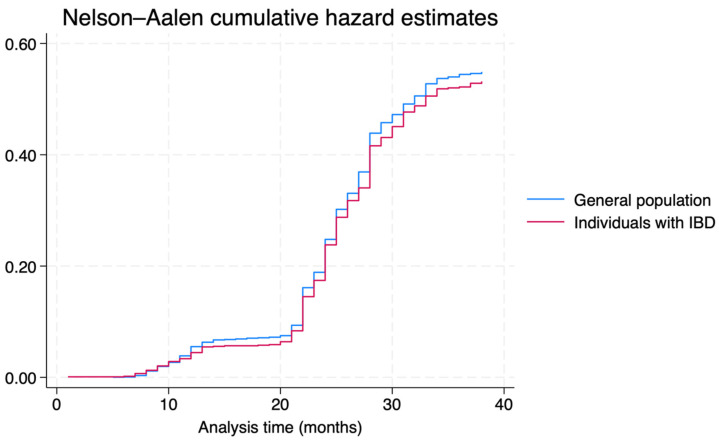
COVID-19 cumulative incidence, per group (Individuals with IBD vs. general population).

**Table 1 vaccines-12-00881-t001:** Characteristics of patients with IBD per group (individuals with IBD vs. general population).

Variable	General Population (*n* = 3075)	Individuals with IBD (*n* = 1029)	Total (*n* = 4140)	*p*-Value
Male; *n* (%)	1685 (54.8)	564 (54.8)	2249 (54.8)	0.994
Age at the start of pandemic; mean ± SD (range)	51.4 ± 16.8 (18–95)	50.8 ± 16.8 (18–96)	51.2 ± 16.8 (18–96)	0.344
Age class; *n* (%) 18–49 50–64 65+	1473 (47.9) 870 (28.3) 732 (23.8)	495 (48.1) 290 (28.2) 244 (23.7)	1968 (48.0) 1160 (28.3) 976 (23.7)	0.994
Comorbidities; *n* (%) none 1 ≥2	1775 (57.7) 868. (28.2) 432 (14.1)	210 (20.4) 603 (58.6) 216 (21.0)	1985 (48.4) 1471 (35.8) 648 (15.8)	<0.0001

**Table 2 vaccines-12-00881-t002:** COVID-19 vaccine coverage per group (individuals with IBD vs. general population).

Variable	General Population	Individuals with IBD	Total	*p*-Value
VC; *n* (%; 95%CI) basal routine first booster second booster	2701 (90.5; 83.9–91.5) 2214 (76.4; 74.8–77.9) 348 (12.2; 11.7–14.3)	918 (91.4; 89.5–93.1) 787 (79.2; 76.5–81.7) 120 (12.2; 10.1–14.4)	3619 (90.7; 89.6–91.6) 3001 (77.1; 75.7–78.4) 468 (12.2; 11.1–13.2)	0.356 0.072 0.992
VC per age class; *n* (%; 95%CI) ● 18–49 years ○ basal routine ○ first booster ○ second booster ● 50–64 years ○ basal routine ○ first booster ○ second booster ● 65+ years ○ basal routine ○ first booster ○ second booster	1285 (87.8; 86.0–89.5) 1000 (68.8; 66.4–71.2) 44 (3.0; 2.2–4.1) 786 (92.9; 90.9–94.5) 661 (81.2; 78.3–83.8) 120 (14.9; 12.5–17.5) 630 (93.1; 90.9–94.9) 553 (87.6; 84.8–90.1) 184 (30.1; 26.5–33.9)	430 (87.2; 83.9–90.0) 357 (72.4; 68.2–76.3) 18 (3.7; 2.2–5.7) 276 (96.8; 94.1–98.5) 241 (84.6; 79.8–88.6) 36 (12.8; 9.1–17.2) 212 (93.8; 89.8–69.6) 189 (87.5; 82.3–91.6) 66 (31.1; 25.0–37.8)	1715 (87.7; 86.1–89.1) 1357 (69.7; 67.6–71.8) 62 (3.2; 2.4–4.1) 1062 (93.9; 92.3–95.2) 902 (82.1; 79.7–84.3) 156 (14.3; 12.3–16.5) 842 (93.2; 91.4–94.8) 742 (87.6; 85.2–89.7) 250 (30.3; 27.2–33.6)	0.721 0.134 0.507 0.016 0.203 0.385 0.698 0.957 0.771
VC per sex; *n* (%; 95%CI) ● Female ○ basal routine ○ first booster ○ second booster ● Male ○ basal routine ○ first booster ○ second booster	1220 (89.8; 88.1–91.4) 1001 (75.7; 73.3–78.0) 138 (10.5; 8.9–12.3) 1481 (91.0; 89.5–92.3) 1213 (77.0; 74.7–79.0) 210 (13.5; 11.9–15.3)	420 (91.1; 88.1–93.5) 361 (79.0; 75.0–82.6) 45 (9.9; 7.3–13.0) 498 (91.7; 89.1–93.9) 426 (79.3; 75.7–82.7) 75 (14.1; 11.2–17.3)	1640 (90.2; 88.7–91.5) 1362 (76.6; 74.5–78.5) 183 (10.4; 9.0–11.9) 1979 (91.2; 89.9–92.3) 1639 (77.6; 75.7–79.3) 285 (13.7; 12.2–15.2)	0.430 0.154 0.718 0.598 0.257 0.754

**Table 3 vaccines-12-00881-t003:** Multivariate logistic regression models of COVID-19 vaccine uptake per dose.

Determinant	Basal Routine		First Booster		Second Booster	
	aOR (95%CI)	*p*-Value	aOR (95%CI)	*p*-Value	aOR (95%CI)	*p*-Value
Group variable (Individuals with IBD vs. General population)	1.06 (1.01–1.03)	0.674	1.00 (0.82–1.22)	0.983	0.90 (0.71–1.15)	0.402
Age (yrs)	1.02 (1.01–1.03)	<0.0001	1.03 (1.02–1.03)	<0.0001	1.07 (1.06–1.07)	<0.0001
Sex (male vs. female)	1.08 (0.87–1.34)	0.478	1.01 (0.86–1.17)	0.934	1.27 (1.03–1.57)	0.025
Comorbidities 1 vs. none ≥2 vs. none	1.20 (0.92–1.56) 1.08 (0.76–1.56)	0.178 0.647	1.43 (1.19–1.72) 1.71 (1.30–2.25)	<0.0001 <0.0001	1.49 (1.16–1.93) 2.37 (1.80–3.14)	0.002 <0.0001
	Goodness-of-fit *p*-value = 0.551		Goodness-of-fit *p*-value = 0.782		Goodness-of-fit *p*-value = 0.096	

**Table 4 vaccines-12-00881-t004:** Multivariate logistic regression models of COVID-19 severe outcomes.

Determinant	Hospitalization		Death	
	aOR (95%CI)	*p*-Value	aOR (95%CI)	*p*-Value
Group variable (Individuals with IBD vs. General population)	0.66 (0.38–1.17)	0.157	0.35 (0.04–2.77)	0.317
Age (yrs)	1.02 (1.01–1.04)	0.003	1.05 (1.01–1.10)	0.030
Sex (male vs. female)	1.74 (1.09–2.79)	0.021	1.14 (0.32–4.08)	0.841
Comorbidities 1 vs. none ≥2 vs. none	1.38 (0.82–2.33) 1.56 (0.84–2.88)	0.225 0.156	0.39 (0.04–3.57) 2.82 (0.70–11.35)	0.401 0.144
Basal vaccination routine	0.37 (0.22–0.62)	<0.0001	0.23 (0.06–0.84)	0.026
	Goodness-of-fit *p*-value = 0.190		Goodness-of-fit *p*-value = 0.999	

## Data Availability

Data are not available due to privacy restrictions.

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
