# Peer review of "Incidence of Severe COVID-19 Outcomes and Immunization Rates in Apulian Individuals with Inflammatory Bowel Disease: A Retrospective Cohort Study"

_vaccines, 2024, doi:10.3390/vaccines12080881_

Round 1

Reviewer 1 Report

Comments and Suggestions for Authors

Review:

COVID19 infects cells through the ACE2, which is present in human intestine epithelial cells and is increased in IBD. So it would be reasonable to think that patients with IBD may be at  a higher risk of having complications when infected. Results around the world are contradictory on this issue. This paper comes to add to the information available. It also considers information on patient vaccination, which is quite relevant.

This is a longitudinal study with a large cohort of patients, the fact that vaccination is also considered, allows to see if vaccines protect IBD patients of the serious outcomes of COVID19.

It is a comprehensive study, where data comes from well established platforms. Also, results could have varied with the population studies, so this adds to existing works.

The statistics methodology used is sound. Patients are 1:3 matched, which gives a strong control.

Conclusions are consistent with the study performed. The study concludes that patients with IBD aren´t at higher risk of severe COVID19 complications. It also performs a multivariate analysis that shows that vaccination is a protective factor, as has been previously shown. It also considers age and sex variations.

References are adequate,

Altogether I consider the paper to be very sound. As I previously mentioned, I would rather have the information shown as charts or bar graphs, since information on tables is harder to visualize.

Author Response

Q1. COVID19 infects cells through the ACE2, which is present in human intestine epithelial cells and is increased in IBD. So it would be reasonable to think that patients with IBD may be at  a higher risk of having complications when infected. Results around the world are contradictory on this issue. This paper comes to add to the information available. It also considers information on patient vaccination, which is quite relevant.

This is a longitudinal study with a large cohort of patients, the fact that vaccination is also considered, allows to see if vaccines protect IBD patients of the serious outcomes of COVID19.

It is a comprehensive study, where data comes from well established platforms. Also, results could have varied with the population studies, so this adds to existing works.

The statistics methodology used is sound. Patients are 1:3 matched, which gives a strong control.

Conclusions are consistent with the study performed. The study concludes that patients with IBD aren´t at higher risk of severe COVID19 complications. It also performs a multivariate analysis that shows that vaccination is a protective factor, as has been previously shown. It also considers age and sex variations.

References are adequate,

Altogether I consider the paper to be very sound. As I previously mentioned, I would rather have the information shown as charts or bar graphs, since information on tables is harder to visualize.

A1. Thank you for your comments. Given the complexity of our arguments, it is challenging to represent the results through graphs. However, we have included additional figures in the Supplementary Appendix.

Reviewer 2 Report

Comments and Suggestions for Authors

The authors present a retrospective cohort study aimed at comparing the incidence of COVID-19 and its outcomes—specifically hospitalization and case-fatality rates—between patients with inflammatory bowel disease (IBD) and the general population in the Apulia region of southern Italy, which has 4,000,000 inhabitants. This study revealed no significant differences in the incidence of COVID-19, the risk of hospitalization, or mortality rates between individuals with IBD and the general population in the Apulia region.

I would like to raise the following concerns.

1.

1)     In Table 1, vaccine coverage is over 90% among IBD patients and 3,075 controls.

2)     Lines 15-16: The study included 1,029 IBD patients and 3,075 controls, matched by age and sex.

3)     Lines 154-156: The group variable (individuals with IBD vs. the general population) was considered the main determinant, adjusted for sex (male vs. female), age (years), and chronic diseases.

4)     Lines 158-159: To analyze the determinants of COVID-19 hospitalization and death, two multivariate logistic regression models were built.

5)     Lines 150-152: The COVID-19 incidence rates (IRs) (per 100 person-months), the proportion of hospitalization, and the case-fatality rate were estimated. The incidence rate ratio (IRR) was calculated to compare the IRs between groups, with 95% confidence intervals (CIs) provided.

Potential Issues with Study Design and Statistical Methods:

1)High Vaccine Coverage:

If the COVID-19 vaccine highly protects individuals from infection, having a sample with more than 90% coverage suggests that the incidence of COVID-19 infection will be very low.

2)Retrospective Cohort Study:

Is it appropriate to use a logistic regression model in a retrospective cohort study, or should a Cox regression model be used instead?

3)Matching Factors in Regression Model:

If age and sex are matching factors, is it appropriate to include them in the regression model?

4)Statistical Methods for Incidence Rates and Ratios:

If incidence rates and incidence rate ratios are calculated, it is suggested to provide the statistical methods used.

2.

In Tables 3-4, comorbidities are included.

This information should be provided in Table 1.

Author Response

Q1. High Vaccine Coverage: If the COVID-19 vaccine highly protects individuals from infection, having a sample with more than 90% coverage suggests that the incidence of COVID-19 infection will be very low.
A1. This is a real-world study. Therefore, it describes what has happened in our population since the pandemic started. The high VC is not a concern, but it is aspected. Considering this important protective factor, the incidence and related complications' values must also be read. The high VC is not a concern for the purpose of our study.

Q2. Retrospective Cohort Study: Is it appropriate to use a logistic regression model in a retrospective cohort study, or should a Cox regression model be used instead?
A2. The choice of the regression model is due to the characteristics of the outcome. We used the logistic model in this case, considering that the outcomes were categorical variables.

Q3. Matching Factors in Regression Model: If age and sex are matching factors, is it appropriate to include them in the regression model?
A3. Considering the study's observational nature and the related risks of bias, we opted to match our groups and adjust the models for age and sex. This is a precaution, especially to reduce the risk of confounding bias.

Q4. Statistical Methods for Incidence Rates and Ratios: If incidence rates and incidence rate ratios are calculated, it is suggested to provide the statistical methods used.
A4. We revised the methods paragraph.

Q5. In Tables 3-4, comorbidities are included. This information should be provided in Table 1.

A5. We revised table 1.

Reviewer 3 Report

Comments and Suggestions for Authors

 The study is interesting and well-written. Few suggestions:

1.    INTRODUCTION: Please in the introduction include a sentence on the importance of immunity on COVID-19 and you may refer to:

a.      Sette A, Crotty S. Immunological memory to SARS-CoV-2 infection and COVID-19 vaccines. Immunol Rev. 2022 Sep;310(1):27-46. doi: 10.1111/imr.13089. Epub 2022 Jun 22. PMID: 35733376; PMCID: PMC9349657.

b.      Petrone L, et al. The Importance of Measuring SARS-CoV-2-Specific T-Cell Responses in an Ongoing Pandemic. Pathogens. 2023 Jun 22;12(7):862. doi: 10.3390/pathogens12070862. PMID: 37513709; PMCID: PMC10385870.

2.    RESULTS:  Is there any possibility to include in your study the immune-modulating therapies provided to individuals with Inflammatory Bowel Disease? These therapies may have an impact on COVID-19 outcomes and response to vaccines (Picchianti-Diamanti A,et al. Immunosuppressive therapies Differently Modulate Humoral- and T-Cell-Specific Responses to COVID-19 mRNA Vaccine in Rheumatoid Arthritis Patients. Front Immunol. 2021 Sep 14;12:740249. doi: 10.3389/fimmu.2021.740249. PMID: 34594343;

3. DISCUSSION: Row 282: after the following sentence “Future scientific studies should address several key areas to gain a more comprehensive understanding of COVID-19 outcomes in patients with IBD..”, please add “as shown in other immune-mediated inflammatory diseases in Picchianti-Diamanti A, et al. Vaccines (Basel). 2023 Nov 2;11(11):1684. doi: 10.3390/vaccines11111684. PMID: 38006015

Author Response

Q1.  INTRODUCTION: Please in the introduction include a sentence on the importance of immunity on COVID-19 and you may refer to:

  1. Sette A, Crotty S. Immunological memory to SARS-CoV-2 infection and COVID-19 vaccines. Immunol Rev. 2022 Sep;310(1):27-46. doi: 10.1111/imr.13089. Epub 2022 Jun 22. PMID: 35733376; PMCID: PMC9349657.
  2. Petrone L, et al.The Importance of Measuring SARS-CoV-2-Specific T-Cell Responses in an Ongoing Pandemic. Pathogens. 2023 Jun 22;12(7):862. doi: 10.3390/pathogens12070862. PMID: 37513709; PMCID: PMC10385870.

A1. We opted to revise the conclusion paragraph with your suggestions.

Q2. RESULTS:  Is there any possibility to include in your study the immune-modulating therapies provided to individuals with Inflammatory Bowel Disease? These therapies may have an impact on COVID-19 outcomes and response to vaccines (Picchianti-Diamanti A,et al. Immunosuppressive therapies Differently Modulate Humoral- and T-Cell-Specific Responses to COVID-19 mRNA Vaccine in Rheumatoid Arthritis Patients. Front Immunol. 2021 Sep 14;12:740249. doi: 10.3389/fimmu.2021.740249. PMID: 34594343;

A2. We agree with your suggestions. Unfortunately, as we already reported in the limitations paragraph, "the primary limitation of our study is that the data sources do not allow for the assessment of specific therapies taken by the IBD population. The scientific literature indicates that the type of therapy is a crucial determinant of severe COVID-19 outcomes. Understanding the impact of various immunosuppressive and biological treatments on COVID-19 severity is vital for providing tailored care to IBD patients. Despite this limitation, it is reasonable to assume that the patients included in our study are under the care of healthcare facilities and, therefore, likely to receive appropriate therapies. Nonetheless, the lack of detailed therapeutic data means our analysis cannot directly correlate treatment types with COVID-19 outcomes.". However, we reported the study you cited in references.

Q3. DISCUSSION: Row 282: after the following sentence “Future scientific studies should address several key areas to gain a more comprehensive understanding of COVID-19 outcomes in patients with IBD..”, please add “as shown in other immune-mediated inflammatory diseases in Picchianti-Diamanti A, et al. Vaccines (Basel). 2023 Nov 2;11(11):1684. doi: 10.3390/vaccines11111684. PMID: 38006015

A3. Revised.

Round 2

Reviewer 2 Report

Comments and Suggestions for Authors

1.

If the COVID-19 vaccine provides high protection against infection, a sample with more than 90% coverage suggests that the incidence of COVID-19 infection will be very low. Therefore, comparing 1,029 IBD patients to 3,075 controls and evaluating outcomes such as hospitalization and death will require sufficient power to determine the relationship accurately.

2.

If the study design is a retrospective cohort study, and the cumulative incidence of infection in the two groups is calculated using Kaplan-Meier curves(log-rank test) and the incidence rate ratio (IRR) between groups, then using a logistic regression model seems inappropriate as it does not account for the effect of the follow-up period.

3.

It is suggested to reassess the suitability of the research design and the statistical analysis methods used in this study. 

Author Response

Q1. If the COVID-19 vaccine provides high protection against infection, a sample with more than 90% coverage suggests that the incidence of COVID-19 infection will be very low. Therefore, comparing 1,029 IBD patients to 3,075 controls and evaluating outcomes such as hospitalization and death will require sufficient power to determine the relationship accurately.

A1. Certainly, we understand your concern regarding the potential impact of the COVID-19 vaccine on our study results. Given that this is an epidemiological study, it is important to clarify that vaccination status was included as a determinant in our regression models to account for its influence. This ensures that the effect of vaccination is appropriately controlled for in our analysis.

However, we acknowledge that the sample size, while adequate for assessing the incidence of COVID-19 infection, may be insufficient to accurately evaluate outcomes such as hospitalization and death due to the relatively low number of these events. We have reported this limitation in our study and, unfortunately, we are unable to expand the sample size further. This constraint should be considered when interpreting the results.

Thank you for your valuable input.

Q2. If the study design is a retrospective cohort study, and the cumulative incidence of infection in the two groups is calculated using Kaplan-Meier curves(log-rank test) and the incidence rate ratio (IRR) between groups, then using a logistic regression model seems inappropriate as it does not account for the effect of the follow-up period.
A2. To investigate the predictors of COVID-19 infection in our sample, we constructed a multivariate semiparametric Cox regression model, which appropriately accounts for the follow-up period and time-to-event data. However, we also utilized logistic regression models because our outcomes were categorical variables. Logistic regression is the most suitable method for analyzing categorical outcomes, allowing us to derive meaningful insights from these variables.

Q3. It is suggested to reassess the suitability of the research design and the statistical analysis methods used in this study. 

A3. Please see above.

Round 3

Reviewer 2 Report

Comments and Suggestions for Authors

No further comment